# Modified Vaccinia Virus Ankara as a Potential Biosafety Level 2 Surrogate for African Swine Fever Virus in Disinfectant Efficacy Tests

**DOI:** 10.3390/pathogens11030320

**Published:** 2022-03-04

**Authors:** Chae Hong Rhee, Moon Her, Wooseog Jeong

**Affiliations:** Veterinary Drugs & Biologics Division, Animal and Plant Quarantine Agency, 177 Hyeoksin 8-ro, Gimcheon-si 39660, Gyeongsangbuk-do, Korea; chrhee82@korea.kr (C.H.R.); herm@korea.kr (M.H.)

**Keywords:** African swine fever virus, disinfectant, modified vaccinia virus Ankara, surrogate virus, virucidal efficacy

## Abstract

In South Korea, despite the increase in emerging viral pathogens in the veterinary industry, only efficacy-tested, virus-specific disinfectants are allowed to be used. Moreover, domestic testing of disinfectants for their virucidal efficacies against foreign, malignant, infectious pathogens that are unreported within the country and/or contagious livestock diseases that require special attention regarding public hygiene are legally restricted. Therefore, the Animal and Plant Quarantine Agency (APQA) designed a study to select a potential biosafety level 2 surrogate of African swine fever virus (ASFV) for efficacy testing to improve the disinfectant approval procedures. For this, the modified vaccinia virus Ankara (MVA) was compared to ASFV in terms of its susceptibility to disinfectants. Effective concentrations of active substances of disinfectants (potassium peroxymonosulfate, sodium dichloroisocyanurate, malic acid, citric acid, glutaraldehyde, and benzalkonium chloride) against ASFV and MVA were compared; similarly, efficacies of APQA-listed commercial disinfectants were examined. Tests were performed according to APQA guidelines, and infectivities of ASFV and MVA were confirmed by hemadsorption and cytopathic effect, respectively. The results reveal that the disinfectants are effective against MVA at similar or higher concentrations than those against ASFV, validating the use of MVA as a potential biosafety level 2 surrogate for ASFV in efficacy testing of veterinary disinfectants.

## 1. Introduction

On 17 September 2019, the Ministry of Agriculture, Food and Rural Affairs (MAFRA) confirmed the first African swine fever (ASF) outbreak in South Korea [1]. The causative pathogen, ASF virus (ASFV), is a large, enveloped double-stranded DNA (170–194 kbp in length) virus of the *Asfarviridae* family [2]. Incidentally, up to 28 January 2022, ASF had been detected in 21 domestic pig farms, with 82 confirmed cases of ASFV-infected pigs [3,4]. According to the Ministry of Environment (ME) [5], if the number of infected wild boars is considered, then there have been 2025 confirmed ASF cases. This is a highly contagious disease that has caused a great concern in the pig industry, and, unfortunately, there are no effective vaccines against it. Therefore, amid this increasing concern of possible virus transmission from wild boars to domestic pigs, Government authorities, i.e., MAFRA along with ME, decided to enforce strengthened control measures, including intensive, nationwide disinfection of farms and livestock-related facilities [5,6].

However, at the time of ASFV emergence, only a few Animal and Plant Quarantine Agency (APQA)-registered disinfectants were available, whose labels specified use against this virus. Therefore, in this emergent situation, APQA classified certain registered disinfectant products based on the presence of active substances with virucidal effects, according to the recommendations of the World Organisation for Animal Health (OIE) and previous studies regarding disinfectants. This enabled the use of the recommended disinfectants against the ASFV, although such applications had not been identified on the original product labels.

In South Korea, virucidal disinfectants, as well as the recommended concentrations at which they should be used against specific viruses, require official approval from APQA, which is provided after assessing the disinfectant according to the APQA guidelines for efficacy testing of veterinary disinfectants [7,8]. However, Article 6 of the current APQA guidelines stipulate domestic test restrictions of disinfectants against foreign, malignant, infectious diseases, such as contagious bovine pleuropneumonia, Rift Valley fever, lumpy skin disease, and ASF, which have not yet been reported in the country [7]. Hence, testing for pathogens that are contagious in livestock or that require special attention with respect to public hygiene must be pre-approved by the commissioner of APQA [7]. Therefore, to acquire approval of disinfectants against ASFV, the disinfectant manufacturers need to entrust the efficacy tests to overseas testing agencies. However, this is problematic for the industry because of the prolonged testing period and increased testing costs. As a result of this regulation-associated situation and the biosafety level (BSL) of ASFV, extensive studies regarding this virus have been hampered. 

Incidentally, ASFV can be manipulated only in high-level containment laboratories, which are not readily available. Nevertheless, this drawback can be circumvented by using a BSL-2 surrogate virus. Surrogates may be defined as organisms, particles, or substances that are used to study the fate of a pathogen in a specific environment [9]. Previously, both pathogenic and nonpathogenic organisms have been employed as surrogates to develop models of such highly infectious pathogens that require high-level containment to be studied [10,11]. Although using the specific target organisms is ideal for any research, the major benefits of using nonpathogenic surrogates are (i) researcher safety, (ii) simple virus propagation in cell cultures, and (iii) the requirement of a lower BSL than that for the actual pathogen.

Although the APQA guidelines have prescribed the use of *Salmonella typhimurium* as a representative bacterium for the bactericidal testing of disinfectants (except for the use of *Clostridium perfringens* and *Mycobaterium fortuitum* for spore-forming and acid-fast bacteria, respectively) [7], there are no specifications regarding a representative test virus for virucidal testing of disinfectants. Therefore, we designed a study for the selection of a representative test virus that can help to improve the performance of the efficiency testing procedure for disinfectants. For this, we planned a two-stage approach: first, research to urgently determine a surrogate virus for ASFV, such that the surrogate may be utilized in efficacy testing of disinfectants against ASFV, and second, perform a simultaneous research to select a representative test virus that encompasses other relevant viruses with respect to disinfectant susceptibility.

The modified vaccinia virus Ankara (MVA) is a large, enveloped double-stranded DNA (178 kbp in length) virus of the *Poxviridae* family, and a highly attenuated strain that has been demonstrated to be safe for humans because of its high safety profile [12]. We examined the MVA and compared it to ASFV in terms of its susceptibility to disinfectants since several previous studies and virucidal efficacy testing guidelines of other countries have extensively studied the MVA and proposed it to be a reference for enveloped viruses [2,12,13,14,15,16,17]. In fact, enveloped viruses, such as ASFV and MVA, are known to be more susceptible to chemical disinfectant than non-enveloped ones [18,19].

Therefore, the present study had two main aims. First, we attempted to evaluate the virucidal efficacy of the representative active substances of the commercial disinfectants used against ASFV against ASFV and MVA. Second, we intended to validate MVA as an appropriate and practical surrogate for virucidal efficacy testing. In conclusion, the virucidal efficacy of the representative active substances and commercial disinfectants against ASFV and MVA were tested following the APQA guidelines. Finally, the observed effective concentrations were compared to evaluate whether MVA would be a suitable surrogate virus for ASFV.

## 2. Results

The concentrations of the tested disinfectants were chosen, based on the formulation of disinfectants that have a proven efficacy against ASFV, to observe the point at which each of the test treatments produced efficient virus inactivation. Subsequently, based on the outcome of the efficacy testing of disinfectant concentrations against ASFV, a sequential change in concentrations was applied in treatments against MVA. A summary of the results for the tested active substances against ASFV and MVA is shown in Table 1 and Table 2. For each active substance, the range of test concentrations includes the recommended concentration for the use of ASFV commercial disinfectants as well as at least one active and one inactive concentration.

A concentration-dependent virucidal activity is observed for all the active substances. The minimal virucidal concentration (MVC) of the active substances against ASFV, which indicates the lowest concentration necessary to inactivate an initial viral titer of ≥4 log, are as follows: potassium peroxymonosulfate (PPMS), 2.5 g/L; sodium dichloroisocyanurate (NaDCC), 1 g/L; glutaraldehyde (GLT), 0.4 g/L; citric acid (CA), 2 g/L; malic acid (MA), 4 g/L; benzalkonium chloride (BZK), 0.1 g/L (Figure 1). None of the concentrations of the active substances exhibits cytotoxicity at 1/10 and 1/100 dilutions.

Similarly, for MVA, the MVCs of the active substances are as follows: PPMS, 2.5 g/L; NaDCC, 1 g/L; GLT, 0.2 g/L; CA, 4 g/L; MA, 4 g/L (Figure 1). However, for BZK, dilutions ≤ 1/100 of the 0.125 g/L treatment are cytotoxic to the BHK-21 cells. The cytotoxic nature of BZK precluded proper detection of infectivity since the decrease in the infectivity titer of 4 log could not be detected. Therefore, the virus inoculum preparation method was modified (undiluted virus inoculum of 1 × 10^9^ TCID_50_/mL) for an overall increase in the viral titers. The consequent MVC of BZK against MVA is 0.25 g/L.

Interestingly, except for GLT, the five other active substances exhibit virucidal activity against MVA at similar or higher concentrations than that against ASFV. The MVC of GLT against MVA is 0.2 g/L; however, this concentration does not exhibit sufficient virucidal efficacy against ASFV, and a ≥ 4 log reduction is obtained with a double GLT concentration (0.4 g/L).

Fortunately, all disinfectant products are confirmed to be effective against ASFV at the authorized concentrations, thereby assuring that the active substances are functional against ASFV at the particular concentrations (Table 3). Thereafter, we applied the disinfectants at these authorized concentrations to test their efficacy against MVA (Table 3). Disinfectants 1, 2, and 3 that contain the effective concentration of PPMS against both ASFV and MVA appear to have a considerable virucidal efficacy against MVA at the authorized concentrations. However, when disinfectants 4 and 5 are used against MVA at the authorized concentrations, their virucidal activities are not enough to meet the 4 log reduction. Furthermore, an adjustment of the dilution of the two disinfectants, 0.2% for disinfectant 4 and 0.4% for disinfectant 5, to reach the effective concentrations of active substances against MVA (0.21 g/L GLT, 0.34 g/L BZK, and 0.16 g/L didecyldimethylammonium chloride (DDAC)) leads to limited determination of virus titer reduction due to cytotoxic reactions. Incidentally, disinfectant 7 is also not effective at the authorized concentration, and dilution of the disinfectant to 2% to reach the effective concentration of the active substance (4.00 g/L CA + 2.00 g/L BZK) leads to cytotoxic reactions. The log reduction in response to disinfectant 6 is limited to 1.20 at the authorized concentration of 0.17%; hence, the dilution of the disinfectant to 1% causes a ≥4.00 log reduction, and a >1% concentration generates ≥4.17 log reduction. Even though, disinfectant 8 does not exhibit the virucidal efficacy necessary to meet the 4 log reduction, a ≥4.00 log reduction is obtained with an increase in the concentration of the disinfectant (2.00 g/L CA and 3.00 g/L sodium lauryl sulfate).

## 3. Discussion

The selection and proper application of disinfectants are important for the control of viral diseases. Therefore, it is necessary to evaluate disinfectants before they are used as a preventive measure. Reportedly, enveloped viruses are more susceptible to chemical disinfectant than non-enveloped ones [18,19]. In fact, ASFV belongs to the category of viruses that is the easiest to inactivate, and all major classes of disinfectants can inactivate it, if used properly. To disinfect the OIE category A viruses that includes enveloped viruses, such as ASFV, the use of detergents, oxidizing agents, alkalis, and glutaraldehyde is recommended [20].

In South Korea, domestic testing of the virucidal efficacy of disinfectants against ASFV has been restricted by the APQA regulations, primarily due to special attention with respect to the public hygiene necessary for investigations associated with the highly contagious ASFV. Hence, to avoid the prolonged test period and increased testing cost of overseas testing services, the use of a surrogate, which can suitably replace ASFV in chemical disinfectant testing, has been suggested.

In this study, the top priority was to efficiently perform the disinfectant approval procedures for ASFV, along with considering a surrogate selection for ASFV. To determine whether MVA is a potential surrogate candidate, according to the APQA testing guidelines, we examined and compared MVA’s response, in terms of susceptibility to disinfectants, to the susceptibility responses of ASFV.

The choice of a surrogate virus is of great importance while establishing a standard test method [15]. Incidentally, MVA has been extensively studied and incorporated into the official virucidal efficacy testing guidelines of other countries as a surrogate for enveloped viruses. In fact, in Europe, the CEN/TC 216 working group 1, which standardizes testing methods and requirements for chemical disinfectants and antiseptics, introduced MVA as a surrogate for all enveloped viruses in 2015. This happened after several studies compared MVA with emerging infectious viruses, such as Ebola virus (EBOV), severe acute respiratory syndrome (SARS-Co-1), and Middle Eastern respiratory syndrome (MERS)-coronavirus (CoV) [14,15,16]. For evaluating the virucidal activity of disinfectants against enveloped viruses in the veterinary field (EN 14675), MVA is being used as a surrogate in the European Standard Guidance on the Biocidal Products Regulation (BPR) [17]. Additionally, the German Veterinary Society (DVG) guidelines have also selected MVA as a surrogate for enveloped viruses during virucidal testing [2].

The virucidal efficacy testing of six active substances revealed that their effective concentrations against MVA are equal to or higher than those against ASFV in five out of six instances. However, the effective concentration of GLT against MVA is 2-fold lower than that against ASFV, thereby implying that MVA is more susceptible to GLT than ASFV. The slight difference in the results may be due to the different approaches that were used in determining the infectivity [2], i.e., HAD for ASFV and CPE for MVA. Brill et al. (2020) examined the virucidal efficacy of GLT, according to EN 17111:2018, and reported that MVA is the most susceptible among four different test viruses (MVA, adenovirus type 5 (AdV), murine norovirus (MNV), and murine parvovirus, minute virus of mice (MVM)) since even 0.1 g/L of GLT is able to achieve a ≥ 4 log reduction [21]. On the contrary, GLT is effective against ASFV at a high concentration (1 g/L), based upon EN 14675 tests [22]. Nevertheless, since most disinfectants are complex formulations of active substances [23], the differences in the efficacies of disinfectant 4 and 5 may be considered insignificant.

Interestingly, our study confirmed that all the eight disinfectants that had been approved for use against ASFV are effective. The concentrations of PPMS in disinfectants 1, 2, and 3, and that of BZK in disinfectants 4, 5, and 6 are within the effective concentration ranges of the respective active substances. Incidentally, BZK belongs to quaternary ammonium compounds (QACs) and DDAC is a comparatively new (fourth generation) QAC with a higher virucidal efficacy than that of BZK [23,24,25]. In disinfectant 7, DDAC is present at 0.17 g/L, and this concentration is sufficient to make the disinfectant effective against ASFV. In disinfectant 8, although the concentration of CA is lower than its effective concentration, this disinfectant is effective due to its formulation with another active substance, namely sodium lauryl sulfate (SLS). Previously, a study reported that the addition of SLS to ineffective formulations improves the virucidal efficacy against only enveloped viruses [26] (p. 189). Hence, it is possible that the synergistic effect of the main substances increases the virucidal efficacy of the entire disinfectant.

The virucidal activities of the officially approved concentrations of ASFV disinfectants against MVA were measured to compare the efficacy of the disinfectants against ASFV and MVA. Disinfectants 1, 2, and 3, which contain 2.00 g/L PPMS as the predominant active substance, are effective against MVA at a concentration equal to the authorized concentration. However, the virus titer reduction is lower than that against ASFV, thereby indicating that MVA is slightly more resistant to disinfection than ASFV. Disinfectants 4, 5, and 7 are thoroughly ineffective against MVA at the authorized concentrations. However, increasing the concentrations of specific active substances to their respective effective concentrations, such as GLT 0.21 g/L for disinfectants 4 and 5 and CA 4.00 g/L for disinfectant 7 induces a cytotoxic effect. Reportedly, commercial disinfectants based on QAC and GLT have high cytotoxicity, thereby limiting the proper assessment of their efficacies [27]. While the dilution of the disinfectant was adjusted to achieve the effective concentration of the targeted active substances, the overall increase in the concentrations of GLT and/or QAC, which had expressed cytotoxicity during individual active substance testing against MVA, seem to cause toxicity of the disinfectants. However, the use of undiluted virus inoculum with high initial titer (modified method application) for an overall increase in viral titers confirmed the efficacy of these disinfectants against MVA, for instance, 6.67 log reduction for disinfectant 4, 6.50 log reduction for disinfectant 5, and 4.33 log reduction for disinfectant 7. Therefore, a higher concentration of disinfectant must be used against MVA, as compared to that used against ASFV, to provide the same or higher disinfectant efficacy. Disinfectants 6 and 8 are not effective against MVA at their respective authorized concentrations. However, upon increasing the specific concentrations of active substances to their respective nearly-effective concentrations, such as CA 3.00 g/L for disinfectant 6 and 2.00 g/L for disinfectant 8, there is a ≥4 log reduction. 

A recent study has reported that the surrogate virus is more resistant to the applied chemical disinfectants than the target outbreak organism [28]. Consistently, our observations depict that MVA is susceptible to the same or higher concentrations of disinfectants, as compared to the susceptibility of ASFV. This information can reinforce the hypothesis that MVA is an appropriate surrogate for ASFV for testing the efficacy of virucidal disinfectants, according to the method described in the APQA guidelines. The use of MVA facilitates the conduction of experiments under BSL-2 facility conditions, thereby avoiding high-level containment areas, which are legally necessary for handling ASFV in South Korea.

This is the first study regarding the virucidal activity testing of representative active substances against ASFV and MVA, according to the APQA guidelines, in South Korea. Based on the results of the virucidal efficacy testing of chemical disinfectants, according to the APQA guidelines, MVA might be an appropriate choice for a BSL-2 surrogate of ASFV.

## 4. Materials and Methods

### 4.1. Disinfectants

Six active substances that represent the main chemical groups included in the formulation of commercially available disinfectants were selected for the study. Their representative products, namely PPMS (Sigma-Aldrich, St. Louis, MO, USA, CAS: 70693-62-8), NaDCC (Sigma-Aldrich, CAS: 2893-78-9), GLT (Grade I, 25% in H_2_O; Sigma-Aldrich, CAS: 111-30-8), CA (Sigma-Aldrich, CAS: 77-92-9), MA (Sigma-Aldrich, CAS: 6915-15-7), and BZK (Sigma-Aldrich, CAS: 63449-41-2), were used. Additionally, eight, APQA-approved, commercial disinfectants, which are currently being used against ASFV, were also examined. All disinfectants were diluted in hard water (CureBio, Seoul, Korea) supplemented with 5% fetal bovine serum (FBS; Corning Inc., Corning, NY, USA) to obtain the specific concentrations that represent high-level organic soiling, as described in the APQA guidelines for efficacy testing of veterinary disinfectants [7,8]. To ensure maximum performance of the disinfectants, the dilutions were prepared immediately before use.

### 4.2. Cells

The BHK-21 cell line (ATCC CCL-10) was purchased from the American Type Culture Collection (ATCC) and routinely maintained in Dulbecco’s modified Eagle’s medium (DMEM; Corning Inc., Corning, NY, USA) supplemented with 10% FBS (Corning Inc.) and 1% antibiotic-antimycotic solution (AA; Corning Inc.). Primary porcine alveolar macrophages (PAMs) were prepared from 11-w-old, specific pathogen-free (SPF) pigs, and these cells were maintained in RPMI 1640 medium (Corning Inc.) supplemented with 10% FBS, 1% AA, and 1% minimum essential medium (MEM) non-essential amino acids (NEAA; Gibco, Grand Island, NY, USA). All cells were grown at 37 °C in a humid atmosphere with 5% CO_2_ saturated with water vapor.

### 4.3. Viruses

The MVA (ATCC VR-1566) was purchased from ATCC. The hemadsorbing ASFV strain Arm/07 (genotype II) (KVCC VR1900047) was provided by the Korean Veterinary Culture Collection (KVCC, Gyeongsangbuk-do, Korea). The MVA was propagated in BHK-21 cells, while the ASFV was propagated in the PAMs. After 5 d of incubation, virus titration was performed by microscopically determining the cytopathic effect (CPE) for MVA-infected BHK-21 cells and hemadsorption (HAD) for ASFV-infected PAMs. The viral titer for MVA was ≥1 × 10^8^ TCID_50_/mL, while that for ASFV was approximately 1 × 10^7^ HAD_50_/mL. Finally, the viruses were stored at −70 °C in 1 mL aliquots until further use.

### 4.4. Virucidal Efficacy Test

The virucidal efficacy tests of the disinfectants were performed in suspension, in accordance with the procedure described by the APQA guidelines for efficacy testing of veterinary disinfectants.

#### 4.4.1. Virus–Disinfectant Reaction

First, the viruses were mixed in the ratio 1:19 with hard water supplemented with 5% FBS to prepare the virus inoculum. Subsequently, the virus inoculum was mixed with each test concentration of the disinfectants in the ratio 1:1 and incubated at 4 °C for 30 min (obligatory condition) [7,8], vortexing every 10 min. For the virus control, hard water was used instead of the disinfectant solution. At the end of reaction time, the ASFV test samples were immediately mixed with an equal amount of RPMI 1640 medium supplemented with 10% FBS, 1% AA, and 1% MEM NEAA, and the MVA test samples were mixed with DMEM supplemented with 10% FBS and 1% AA to quench the chemical activity of disinfectants. Subsequently, tenfold serial dilutions (up to 10^−7^) were performed in the 2% FBS-supplemented medium of each virus. It was essential that the virus titer in the virus control was at least 2 × 10^5^ HAD_50_/mL or TCID_50_/mL. The infectivity was determined using end-point dilution titration in microtiter plates.

#### 4.4.2. HAD Assay for ASFV

First, 100 µL of PAMs (1 × 10^6^ cells/mL) were seeded in each well of a 96-well microtiter plate. The medium was removed after 24 h of incubation at 37 °C in a CO_2_ incubator. Thereafter, 100 µL of the prepared dilutions of the virus–disinfectant mixtures were inoculated in each well. After 1 h, 30 µL of 0.04% pig erythrocytes was added to every well. These treatment set-ups were observed every day, and microscopic examination confirmed the occurrence of HAD after 5 d [29,30]. The viral titer was calculated as 50% hemadsorbing dose per mL (HAD_50_/mL) in log units using the Spearman–Karber method [31]. All experiments using ASFV were performed in a BSL-3 facility at the APQA in South Korea.

#### 4.4.3. Tissue Culture Infectious Dose Assay for MVA

Aliquots of 100 µL from each dilution of the virus–disinfectant mixtures were transferred into 6 wells of a 96-well microtiter plate, each containing 50 µL of BHK-21 cell cultures (1 × 10^6^ cells/mL), followed by incubation at 37 °C in a CO_2_ incubator. The treatment set-ups were observed every day, and microscopic examination confirmed the incidence of virus-induced CPE after 5 d. The viral titer was calculated as 50% tissue culture infectious dose per mL (TCID_50_/mL) in log units using the Spearman–Karber method [31]. All experiments using MVA were performed in a BSL-2 facility.

### 4.5. Cytotoxicity Test

Preliminary studies were performed to determine potential cytotoxicity of the chemical disinfectants to the cells. To evaluate the chemical-induced cytotoxicity of the disinfectants, hard water was used to prepare the inoculum instead of the virus. The cells were treated in the same manner as described above. Hard water was mixed with each test concentration of the disinfectants in the ratio 1:1 and incubated at 4 °C for 30 min, vortexing every 10 min. At the end of reaction time, the test samples were immediately mixed with an equal amount of RPMI 1640 medium supplemented with 10% FBS, 1% AA, and 1% MEM NEAA or DMEM supplemented with 10% FBS and 1% AA to quench the chemical activity of disinfectants. Subsequently, serial tenfold dilution (up to 10^−2^) was performed in the 2% FBS-supplemented medium of each cell, and aliquots of 100 µL from each dilution of the test mixtures were transferred into 6 wells of a 96-well microtiter plate, each containing cell cultures (1 × 10^6^ cells/mL), followed by incubation at 37 °C for 5 d in a CO_2_ incubator. The cytotoxicity control wells were examined for any cell death due to the presence of residual toxicity from the chemical used. Results were considered acceptable if no cytotoxicity was observed at a 1:10 dilution.

### 4.6. Data Analysis

For each concentration of the disinfectant–virus mixtures, experiments were performed in triplicates. Inactivation efficacy against the viruses was expressed as the titer reduction in log units, presented as the difference between the virus titer after reaction with disinfectant and the control virus titer. According to the APQA guidelines, the final virus titer was determined as the median value of the triplicates within a 20% (±10%) error range, and the virus inactivation was considered to be effective when the titer reduction was ≥4 log [7]. An average of three reduction values was calculated, and the standard deviation and median values were determined.

## Figures and Tables

**Figure 1 pathogens-11-00320-f001:**
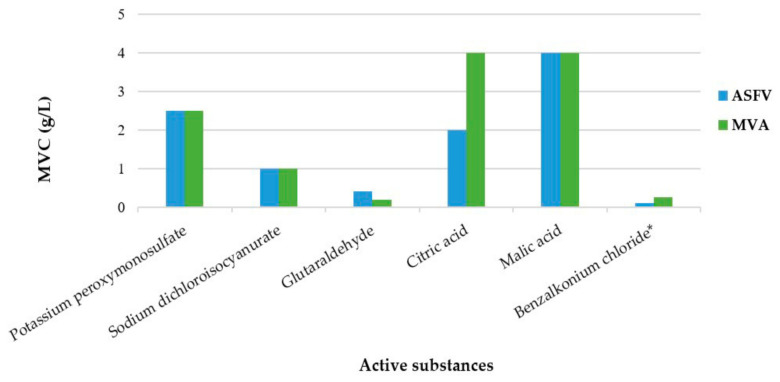
Minimal virucidal concentrations (MVC) of the representative active substances against African swine fever virus (ASFV) and modified vaccinia virus Ankara (MVA). MVC indicates the lowest concentration necessary to inactivate an initial viral titer of ≥4 log. * Limitation encountered during determination owing to cytotoxicity.

**Table 1 pathogens-11-00320-t001:** Virucidal activity of active substances against African swine fever virus.

Disinfectant (Active Substance)	Concentration (g/L)	Log_10_ Reduction				
		E1	E2	E3	Mean (±SD)	Median
Potassium peroxymonosulfate	1.25	2.17	2.50	1.83	2.17 ± 0.34	2.17
	2.5	**5.83**	**4.33**	**5.17**	**5.11 ± 0.75**	**5.17**
Sodium dichloroisocyanurate	0.5	2.17	2.83	2.17	2.39 ± 0.38	2.17
	1	**5.17**	**5.33**	**4.50**	**5.00 ± 0.44**	**5.17**
Glutaraldehyde	0.2	2.67	2.67	2.33	2.56 ± 0.20	2.67
	0.4	**4.17**	**4.00**	**4.67**	**4.28 ± 0.35**	**4.17**
Citric acid	1	2.50	3.33	2.50	2.78 ± 0.48	2.50
	2	**5.00**	**4.83**	**4.67**	**4.84 ± 0.15**	**4.83**
Malic acid	2	3.50	3.33	3.17	3.33 ± 0.17	3.33
	4	**4.33**	**5.33**	**4.17**	**4.61 ± 0.63**	**4.33**
Benzalkonium chloride	0.05	2.00	2.83	2.33	2.39 ± 0.42	2.33
	0.1	**4.50**	**4.83**	**4.50**	**4.61 ± 0.19**	**4.50**
	0.2	**4.50**	**4.83**	**5.16**	**4.83 ± 0.33**	**4.83**

Values in bold indicate effective virucidal activity (viral titer ≥ 4 log_10_). E, experiment; SD, standard deviation.

**Table 2 pathogens-11-00320-t002:** Virucidal activity of active substances against modified vaccinia virus Ankara.

Disinfectant (Active Substance)	Concentration (g/L)	Log_10_ Reduction				
		E1	E2	E3	Mean (±SD)	Median
Potassium peroxymonosulfate	1.25	3.50	**4.00**	3.00	3.50 ± 0.50	3.50
	2.5	**4.17**	**4.33**	**4.50**	**4.33 ± 0.17**	**4.33**
	5	**5.00**	**4.50**	**5.00**	**4.83 ± 0.29**	**5.00**
Sodium dichloroisocyanurate	0.5	2.17	1.67	0.67	1.50 ± 0.76	1.67
	1	**5.00**	**4.50**	**5.00**	**4.83 ± 0.29**	**5.00**
	2	**5.00**	**4.50**	**5.00**	**4.83 ± 0.29**	**5.00**
Glutaraldehyde	0.1	3.67	3.50	3.83	3.67 ± 0.17	3.67
	0.2	**4.50**	**4.67**	**4.50**	**4.56 ± 0.10**	**4.50**
Citric acid	2	2.00	0.17	1.67	1.28 ± 0.98	1.67
	4	**4.50**	**4.00**	**4.17**	**4.22 ± 0.25**	**4.17**
Malic acid	2	2.33	0.83	2.00	1.72 ± 0.79	2.00
	4	**4.33**	**4.50**	**4.50**	**4.44 ± 0.10**	**4.50**
Benzalkonium chloride	0.0625	0.67	0.83	0.67	0.72 ± 0.09	0.67
	0.125 *	1.50	1.17	1.33	1.33 ± 0.17	1.33
	0.25 *	**5.50**	**5.33**	**4.50**	**5.11 ± 0.54**	**5.33**

Values in bold indicate effective virucidal activity (viral titer ≥ 4 log_10_). E, experiment; SD, standard deviation. * Cytotoxicity.

**Table 3 pathogens-11-00320-t003:** Description and virucidal activity of eight commercial disinfectants tested against African swine fever virus and modified vaccinia virus Ankara.

Disinfectant	Active Substance	Concentration of Disinfectant (%)	Active Substance Concentration (s) within the Concentration of Disinfectant Use (g/L)	Log_10_ Reduction	
				ASFV	MVA
1	PPMS + NaDCC	**0.40**	2.00 + 0.20	4.67	4.00
2	PPMS + MA	**0.50**	2.50 + 0.50	6.17	4.00
3	PPMS + NaDCC	**0.50**	2.50 + 0.25	4.67	4.20
4	GLT + QAC (BZK+DDAC)	**0.17**	0.18 + 0.41 (BZK 0.28 + DDAC 0.13)	4.17	2.67
		0.20	0.21 + 0.50 (BZK 0.34 + DDAC 0.16)	n.d.	≥3.30 *
5	GLT + QAC (BZK+DDAC)	**0.33**	0.18 + 0.41 (BZK 0.28 + DDAC 0.13)	4.17	3.17
		0.40	0.21 + 0.50 (BZK 0.34 + DDAC 0.16)	n.d.	≥3.00 *
6	CA + QAC (BZK)	**0.17**	0.50 + 0.17	5.17	1.20
		0.33	1.00 + 0.33	n.d.	1.83
		0.50	1.50 + 0.50	n.d.	3.67
		1.00	3.00 + 1.00	n.d.	4.00
		1.33	4.00 + 1.33	n.d.	4.17
7	CA + QAC (DDAC)	**0.17**	0.33 + 0.17	5.17	1.80
		2.00	4.00 + 2.00	n.d.	≥2.20 *
8	CA + SLS	**0.26**	1.05 + 1.58	5.00	3.70
		0.33	1.33 + 2.00	n.d.	3.80
		0.50	2.00 + 3.00	n.d.	4.00
		1.00	4.00 + 6.00	n.d.	4.70

PPMS, potassium peroxymonosulfate; NaDCC, sodium dichloroisocyanurate; GLT, glutaraldehyde; CA, citric acid; MA, malic acid; BZK, benzalkonium chloride; QAC, quaternary ammonium chloride; DDAC, didecyldimethylammonium chloride; SLS, sodium lauryl sulfate. ASFV, African swine fever virus; MVA, modified vaccinia virus Ankara; n.d., not done. Bold type, authorized use concentration against African swine fever virus. * Limitation encountered during determination owing to cytotoxicity.

## Data Availability

Not applicable.

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
