# Peer review of "Modified Vaccinia Virus Ankara as a Potential Biosafety Level 2 Surrogate for African Swine Fever Virus in Disinfectant Efficacy Tests"

_pathogens, 2022, doi:10.3390/pathogens11030320_

Round 1
Reviewer 1 Report
ASF is an economically important infectious disease for global pig industry. Up to date, the commercial vaccine is still unavailable, and any treatment is forbidden, therefore the only way to control the disease is provided by strict sanitary measures, including efficient disinfection. The manuscript entitled “Modified vaccinia virus Ankara as a potential, biosafety level-2 2 surrogate for African swine fever virus in disinfectant efficacy 3 tests“ describes comparison of ASFV sensitivity to disinfection with MVA. The article contains comparisons of two viruses, but at the outset does not mention their structure and the related sensitivity of viruses to disinfectants. Manuscript is well written, introduction needs refilling, methods are adequate. However some minor issues must be improved:
Minor comments:
The introduction should include a comparison of the structure and sensitivity of both ASFV and MVA viruses
Line 32, 37: Authors didn’t make the abbreviation regarding the Ministry of the Environment, and then in line 37 they used the abbreviation ME, probably about this Ministry
Line 87-89: Authors didn’t specify which virus this study should be performed on
Line 104-110: There is inconsistency in the amount of thb chloride against ASFV and MVA. In the case of tests against ASFV, 2 concentrations of the above substances were performed, while against MVA – 3
Line 262-… “Materials and methods” section should be right below the introduction so that the reader understands how the results were obtained
Line 331-334: “Cytotoxicity test” section is very brief, it doesn’t sufficiently describe the method of eliminating cytotoxicity, and cytotoxicity cann’t be synonymous with the effectiveness of the agent, because cells are destroyed.
Author Response
"Please see the attachment."

Reviewer 2 Report
The manuscript performed by Jeong et al to prove that the APQA-listed commercial disinfectants are effective against MVA at similar or higher concentrations than that against ASFV. The research is interesting and the results are convincing.
Major concerns:
1# the APQA guidelines have prescribed the use of Salmonella Typhimurium as a representative bacterium for the bactericidal testing of disinfectants (except for the use of Clostridium perfringens and Mycobaterium fortuitum for spore-forming and acid-fast bacteria respectively), more tests might be needed for virucidal testing of the disinfectants on viruses.
2# A surrogate virus is of great importance while establishing a test method, while the real efficacy need more tests and discussion.
Author Response
"Please see the attachment."
